# Nanocomposites of Rigid Polyurethane Foam and Graphene Nanoplates Obtained by Exfoliation of Natural Graphite in Polymeric 4,4′-Diphenylmethane Diisocyanate

**DOI:** 10.3390/nano12040685

**Published:** 2022-02-18

**Authors:** Se-Ra Shin, Dai-Soo Lee

**Affiliations:** 1Research Institute, Jung-Woo Fine Corp., Ltd., 63-8, Seogam-ro 1-gil, Iksan 54586, Korea; srshin89@jbnu.ac.kr; 2Division of Semiconductor and Chemical Engineering, Jeonbuk National University, 567 Baekjedaero, Deokjin-gu, Jeonju 54896, Korea

**Keywords:** graphene, liquid crystalline, rigid polyurethane foam, nanocomposites, thermal insulation

## Abstract

The influence of graphene nanoplates (GNPs) obtained by the ecofriendly exfoliation of natural graphite has been addressed on the mechanical and thermal insulating properties of rigid polyurethane foams (RPUFs). Few-layer GNPs with few defects were prepared in polymeric 4,4′-diphenylmethane diisocyanate (pMDI) under ultrasonication to obtain a GNP/pMDI dispersion. GNP/pMDI dispersions with different GNP concentrations were used to prepare RPUF nanocomposites via in situ polymerization. An important finding is that the GNP/pMDI dispersion exhibits lyotropic liquid crystalline behavior. It was found that the unique orientation of GNPs above the concentration of 0.1 wt% in the dispersion affected the mechanical and thermal insulation properties of the RPUF nanocomposites. GNP/RPUF nanocomposites with GNP concentrations at 0.2 wt% or more showed better thermal insulating properties than neat RPUF. The lyotropic liquid crystalline ordering of GNPs provides stable nucleation for bubble formation during foaming and prevents bubble coalescence. This decreases the average cell size and increases the closed cell content, producing GNP/RPUF nanocomposites with low thermal conductivity. Furthermore, GNPs incorporated into RPUF act as a barrier to radiant heat transfer through the cells, which effectively reduces the thermal conductivity of the resulting nanocomposites. It is expected that the nanocomposite of RPUF investigated in this study can be applied practically to improve the performance of thermal insulation foams.

## 1. Introduction

Polyurethane (PU) foams are cellular polymers comprising a solid polymer matrix phase and a gas phase formed via the physical and chemical reactions of a blowing agent [1,2]. Depending mainly on the rigidity of cellular polymers, polyurethane foams are classified as rigid, semi-rigid, or flexible. Among these, rigid polyurethane foams (RPUFs) with closed cellular structures are generally used for thermal insulation in buildings and refrigerators due to their low thermal conductivity, high strength-to-weight ratios, easy processing, and low density [1,2,3]. Thermal conductivity is one of the most important properties for insulation, and this is strongly affected by various parameters, including density, cell size, closed cell content, and the thermal conductivity of blowing gas trapped in closed cells [1,4,5,6]. The thermal conductivity of the RPUFs can be expressed as the sum of several contributions, as follows [7]:k_t_ = k_s_ + k_g_ + k_r_(1)
where k_t_ is the total thermal conductivity of the foam, and k_s_, k_g_, and k_r_ are the thermal conductivities of the solid phase, the blowing gas, and the radiation term across the cells, respectively. The thermal conductivities of the solid and gas phases can be regarded as constants, as these are inherent to the raw materials. Thus, minimizing the radiative thermal conductivity is an important factor with respect to improving the thermal insulation performance of RPUFs. The radiative thermal conductivity term (k_r_) can be decreased by the presence of nanofillers, which reduce the cell size and radiative heat transfer [7,8,9,10,11,12].

In past decades, the enhancement of the mechanical and thermal insulation properties of RPUF by incorporation of various nanofillers, such as graphene [3,13,14,15,16,17], nanoclay [4,8,12], cellulose [11,18], carbon nanofibers [8,19], and carbon nanotubes [3,20] have often been reported. Nanofillers act as nucleating agents to reduce the cell size and improve the thermal insulation properties of an RPUF. The mechanical properties of the filled compositions may also be enhanced due to the stiffness of the nanofillers present. For example, RPUF nanocomposites reinforced with 3 wt% organoclay [4] have been prepared. The organoclay was dispersed in pMDI using mechanical stirring, followed by ultrasonication to achieve a uniform dispersion. Applying ultrasonication to disperse the organoclay could effectively be used to generate a RPUF composite with tensile strength and thermal insulation properties greater than those of unmodified RPUF. The organoclay might function as a nucleating agent for bubble growth during foaming, resulting in a reduced cell size and fine cell morphology. RPUF nanocomposites containing a low level (~0.3%) of GNPs or carbon nanotubes (CNTs) [3] had been prepared. Incorporating GNPs and CNTs into the resin enhanced the mechanical properties and heat resistance while hardly changing the thermal conductivity. The presence of GNPs led to superior enhancement of mechanical and thermal properties compared with those of resins containing CNTs, which may be attributed to the strong interfacial adhesion between GNPs and the polyurethane matrices. RPUFs containing a small quantity of graphene (0.3 and 0.5 wt%) [9] have been generated. RPUF nanocomposites were prepared using commercially available graphene of 1~2 layers. The radiative thermal conductivity of the modified RPUF was lower than that of RPUF alone. The initial thermal conductivity and thermal aging rate were reduced by loading 0.3 wt% graphene. The improvement in the thermal insulation properties was attributed to a reduction in the radiative contribution to thermal conductivity resulting from a decreased cell size and an increased extinction coefficient. The foaming of polyurethane with a 1 wt% carbon nanofiber (CNF) dispersion in polyol or polyisocyanate has been studied [19]. An effective reduction in the thermal conductivity and enhancement of the normalized compressive modulus were observed for RPUF composites with a minimal CNF loading and no structural defects. Nucleation by CNF led to the development of more uniform cell morphology, and an improvement in the physical properties of the RPUF nanocomposites.

Among the carbonaceous nanofillers mentioned above, graphene is an allotrope of carbon with a two-dimensional lattice. Its outstanding thermal, mechanical, and electrical properties and high electron mobility allow its use in many applications, including the production of sensors, electronic devices, polymeric materials, fibers, and liquid crystal devices [21,22,23,24,25,26,27,28,29,30]. In particular, graphene is an excellent reinforcement agent for improving the thermal and mechanical properties of polymer composites even at very low concentrations due to its unique two-dimensional structure and high specific surface area. Graphene nanoplates (GNPs) are generally obtained by the chemical or thermal oxidation of exfoliated graphene oxide (GO) [31,32]. The presence of various functional groups in the basal plane of GO modifies the inherent properties of graphene by disrupting its conjugated structure [32]. These functional groups can be removed by chemical or thermal reduction under harsh conditions, resulting in conversion to reduced graphene oxide [32,33]. However, a considerable number of structural defects remain in its basal plane after reduction [32,34]. To overcome this problem, direct exfoliation of natural graphite to generate graphene has been conducted in various organic solvents [31,35], ionic liquids [36], surfactants [37], and polymer matrices under ultrasonication [38,39]. However, liquid phase exfoliation using organic solvents induces aggregation via the restacking of graphene sheets due to strong van der Waals interactions during the removal of organic solvents. Although exfoliation by ionic or non-ionic surfactants inhibits the restacking of graphene sheets, incomplete removal of surfactants may result in deterioration of the specific properties of polymer composites because of reduced interfacial adhesion [40,41]. Therefore, a low viscosity polymer might be a suitable dispersion medium for the direct liquid phase exfoliation of natural graphite by employing in-situ polymerization. The polymer medium intercalated between graphene layers may develop sufficiently stable graphene dispersions to prevent aggregation; this technique can be applied to polymer composite preparation via in situ polymerization [26,42,43].

GNPs with few defects have been prepared using a simple and eco-friendly process without solvents under ultrasonication to obtain a GNP/pMDI dispersion. This represents the first attempt to obtain GNP dispersions in pMDI by the exfoliation of natural graphite (NG) in the absence of organic solvents, and the attempt can be employed practically without difficulties. Here, pMDI was used as a medium for exfoliating NG, and adsorption of pMDI on the GNP surface prevented restacking of the GNPs. Interestingly, the GNP/pMDI dispersion displays a unique GNP orientation depending on the GNP concentration, e.g., lyotropic liquid crystalline behavior in rheological measurements. GNP/pMDI dispersions with different GNP content were used to prepare RPUF nanocomposites via in situ polymerization. The effects of the GNP orientation on the thermal insulation properties, as well as the thermal and mechanical properties, of the resulting nanocomposites have been assessed.

## 2. Materials and Methods

### 2.1. Materials

Fifty-mesh NG was purchased from Hyundai Coma Industry Inc. (Seoul, Korea). An aromatic amine-based polyol with a hydroxyl number of 309.2 mg KOH/g was a product of Jung-Woo Fine Chemical Ltd. (Ik-san, Korea). Polymeric diphenylmethane diisocyanate (pMDI), with a functionality of 2.7 and NCO content of about 31%, was supplied by Kumho Mitsui Chemicals (Yeo-su, Korea). Catalysts such as dimethyl cyclohexylamine (DMCHA, Polycat^®^ 8, PC-8) and potassium 2-ethylhexanoate dissolved in diethylen glycol (PEH, potassium HEX CEM-977) were obtained from Air Products and Chemicals (Allentown, PA, USA) and OMG (Cleveland, OH, USA), respectively. Silicon surfactant (B-8462) and tris(1-chloro-2-propyl) phosphate (TCPP) were purchased from Evonik (Essen, Germany) and Aldrich (Yong-in, Korea), respectively. SOLKANE^®^ 365/227 from Solvay Special Chemicals (Brussel, Belgium) was used as a physical blowing agent, and distilled water was used as a chemical blowing agent. All chemicals were used as received.

### 2.2. Preparation of GNP/pMDI Dispersion

Exfoliated graphite was obtained by ultrasonicating NG in pMDI as a medium without solvent. In order to determine the concentration of NG in pMDI to perform the sonication to get GNP from NG, concentrations of NG in pMDI for the sonication were varied from 0.4 to 4 phr in preliminary experiments. It was found that the yields of GNP were ca 50% and changed little. Thus, 2 phr of NG in pMDI was exfoliated by ultrasonication at 400 W for 26 h in this study. Further increase in the time of ultrasonication did not improve the yield of exfoliated graphite. The graphite/pMDI mixture was then centrifuged at 3000 rpm for 20 min to yield exfoliated graphite with a few layers. The obtained supernatant was poured into a glass bottle sealed with nitrogen. Finally, the GNP/pMDI dispersion was obtained, containing a GNP concentration of 1.0 wt%, corresponding to a 50% exfoliation yield. The highly concentrated GNP/pMDI dispersion was diluted by mixing with additional pMDI to obtain GNP/pMDI dispersions with different GNP concentrations. The process for preparing the GNP/pMDI dispersion is illustrated in Figure 1.

### 2.3. Preparation of RPUF Nanocomposites

RPUF/GNP nanocomposites were prepared via a one-shot method. Initially, fixed amounts of surfactant, catalyst, phosphate, and blowing agents were added to the polyol in a polypropylene beaker and mixed until they became homogeneous. Predetermined amounts of GNP/pMDI dispersions with different GNP contents (0.1, 0.2, 0.3, and 0.4 wt% in pMDI) were then added to the polyol mixture and mixed vigorously at 6000 rpm for 7 s. The mixture was poured into a closed steel mold (300 × 300 × 50 mm^3^) with a lid. After curing at 60 °C for 20 min, the RPUF/GNP nanocomposites were demolded and cured for at least 24 h at room temperature before conducting measurements. The NCO index was maintained at 120%. The sample codes and formulations of the RPUF nanocomposites are summarized in Table 1. Control RPUF was prepared for comparison with the RPUF nanocomposites, using the same procedure as for the preparation of GNP-containing RPUF nanocomposites.

### 2.4. Characterization

Raman spectroscopy (RAMAN, Nanofinder 30, Tokyo Instruments, Tokyo, Japan) was performed with a He-Ne laser at 633 nm. High-resolution transmission electron microscopy (HR-TEM, JEM-2010, JEOL, Akishima, Japan) was employed to observe the morphology of GNPs placed on a carbon TEM grid. Atomic force microscopy (AFM, Nanoscope III, Veeco Instruments, Plainview, NY, USA) was employed in tapping mode to determine the thickness of GNPs placed on a silicon wafer. The silicon wafer was washed three times with ethanol under ultrasonication before use. The rheological behavior of the GNP/pMDI dispersion was studied using a rheometer (AR-2000, TA Instruments, New Castle, DE, USA) equipped with a parallel plate in steady shear rotation mode at room temperature. The reactivity of RPUF nanocomposites during foaming and polymerization were characterized by cream, gel, and tack free times according to ASTM D7487-13. The compressive strengths of the RPUF nanocomposites were determined with a universal testing machine according to ASTM D 1621 with a sample size of 40 × 40 × 40 mm^3^. The compressive strengths were measured for five specimens per sample, and the average values were reported. The measured compressive strengths were normalized by a density of about 42 ± 1 kg/m^3^ to eliminate density effects. The thermal conductivities of the RPUF nanocomposites were obtained using a heat flow meter (HFM 436 Lambda, Netzsch, Selb, Germany) with a two-plane plate maintained at different temperatures, according to ASTM C 518. The thermal conductivities were obtained for three specimens per sample, and the average values were reported. The error bars are given in the Figures to show the experimental uncertainty. However, they are not seen in Figures when the standard errors of the data were smaller than the size of symbols for the data.

The closed cell contents of the samples were measured using an ULTRAPYC 1200e from Quantachrome (Boynton Beach, FL, USA), according to ASTM D 6226, with sample dimensions of 25 × 25 × 25 mm^3^. The closed cell contents were averaged for five specimens per sample. The cell morphology of the nanocomposites was observed by scanning electron microscopy (JSM-6400, JEOL Ltd., Akishima, Tokyo, Japan) at an accelerating voltage of 20 kV. The samples were prepared by a cryogenic fracture technique and coated with gold before observation. The dynamic mechanical properties of RPUF nanocomposites were studied using a dynamic mechanical analyzer (Q800, TA Instruments, New Castle, DE, USA). The measurements were performed in tension mode from 30 to 250 °C at a heating rate of 3 °C/min (at frequency 1 Hz and amplitude of 15%).

## 3. Results & Discussion

### 3.1. Characteristics of Exfoliated Graphite

Exfoliated graphite was successfully obtained by ultrasonicating NG in pMDI, which functioned as a dispersion medium. Few-layer GNPs with few defects were obtained, and the GNP/pMDI dispersion obtained was conveniently used to prepare the RPUF nanocomposites in this study. For the Raman, TEM, and AFM measurements, GNPs without pMDI were obtained from the GNP/pMDI dispersion by microfiltration and washing with acetone. Raman spectroscopy is widely used to investigate the degree of exfoliation and structural defects of nanoscale graphene with unique structure and properties [44,45]. Figure 2 presents the Raman spectra of GNPs exfoliated with pMDI. Three characteristic peaks were observed for both GNPs and NG. The G peak at around ~1580 cm^−1^ mainly originates from the vibration of *sp^2^* carbon atoms with adjacent atoms in their structure, and the D peak at ~1360 cm^−1^ originates from defects and disorder in the graphitic structure. The D peaks seen in NG and GNPs are not usually observed in highly oriented pyrolytic graphite. Typically, the intensity ratio of D to G peaks (*I*_D_/*I*_G_) is used as an indicator of the proportion of defects on the graphene surface. In general, *I*_D_/*I*_G_ tends to increase with the proportion of defects or oxidation level of graphitic materials. The ratios of *I*_D_/*I*_G_ for the NG and GNPs were 0.1 and 0.2, respectively. The increase of *I*_D_/*I*_G_ in the GNPs compared with NG is attributable to defects introduced by the ultrasound sonication treatment. The observed ratios of *I*_D_/*I*_G_ for GNPs are comparable to those reported in previous studies [46,47,48,49,50]. The 2D peak of graphitic materials in Raman spectra (2700 cm^−1^) is related to their stacking order, providing information on the approximate number of layers of graphene sheets [44,45]. The 2D peak of the GNPs (2666 cm^−1^) was shifted slightly to the left (blue shift) compared with that of NG (2679 cm^−1^), indicating that the exfoliated GNPs had approximately 4–6 layers [45].

TEM and AFM analyses were performed to determine the precise characteristics of the GNPs, and the results are presented in Figure 3 and Figure 4. Figure 3 presents an AFM image and a line profile of GNPs. The green triangle points indicate the height difference between the GNP surface and the residual pMDI that was not completely removed, and the red triangle points in the AFM image indicate the thickness of the GNPs. It can be seen that the GNPs had a thickness of about 1.93 nm. Figure 4a shows that the GNPs had a sheet-like morphology with average sizes of 0.2~0.5 µm. However, some aggregation and restacking of the GNP layers could be observed due to removal of pMDI, preventing the aggregation of GNPs by van der Waals forces. Figure 4b presents a folded edge image of a GNP. A GNP consists of around six layers, and, considering that the thickness of monolayer graphene is ~0.34 nm, which is consistent with a thickness of 2 nm. A selected area electron diffraction (SAED) pattern, used to study the crystallinity of nanoscale materials, is also presented at the bottom right of Figure 4b. The SAED pattern of the GNPs showed well-defined diffraction points, matching the hexagonal lattice. Thus, these results demonstrated that few-layer GNPs with few defects can be produced by the ultrasonic exfoliation of NG in pMDI in the absence of solvent.

### 3.2. Characteristics of GNP/pMDI Nanodispersions

The viscosity of a normal polymer solution or colloidal dispersion generally increases with concentration. However, in the present study, the steady shear viscosity showed unusual lyotropic liquid crystalline behavior with respect to the GNP concentration of the GNP/pMDI dispersion, as presented in Figure 5. The shear viscosity increased with the GNP content until it reached 0.1 wt%. In this range, GNPs are randomly dispersed in pMDI. Subsequently, the viscosity of dispersion was slightly decreased, and reached a minimum at 0.2 wt% of GNP content, where the random dispersion and parallel alignment of GNP coexisted. With a further increase in GNP content, the viscosity of GNP/pMDI dispersions increased and it exhibited the denser parallel alignment of GNP. The non-monotonic behavior near the critical GNP volume fraction was due to the transition from the isotropic to the nematic liquid crystalline phase [51]. It was found that the GNP/pMDI dispersion showed the typical viscosity change of lyotropic liquid crystalline solutions [51,52].

### 3.3. Characteristics of GNP/RPUF Nanocomposites

In this study, RPUF nanocomposites with a density of around 42 ± 1 kg m^−3^ were prepared from GNP/pMDI dispersions. The reactivities with regard to blowing and polymerization in preparing the RPUF nanocomposites, including cream time (CT), gel time (GT), and tack free time (TFT), are presented in Figure 6. CT is the point of color change from dark yellow to bright cream due to bubble formation. GT is the starting point of gelling and cross-linking by formation of urethane, urea, and allophanate linkages, and TFT is the point when the foam surface loses its stickiness, allowing the foam to be demolded. No significant changes in reactivities were observed for any of the tested RPUF nanocomposites, indicating that adding GNPs did not affect the reactivity in preparing the RPUF nanocomposites.

In this study, RPUFs were prepared from polyols and the GNP/pMDI dispersion obtained by the process shown schematically in Figure 1. It was confirmed by the AFM image and TEM images of the GNPs in the GNP/pMDI dispersion that the thickness of the GNPs was ca 2 nm (Figure 4). Thus, RPUFs prepared from the GNP/pMDI were described to be nanocomposites as they were obtained by the fast reaction of polyols and pMDI containing GNP in colloidal state in which, gel time was ~40 s and the agglomeration of GNP could not possibly occur. However, further studies on the dispersion of GNPs in RPUF are necessary.

Figure 7 presents the cell morphologies of RPUF nanocomposites reinforced with GNPs. The average cell size and the number of cells per unit area (cm^2^) of RPUF nanocomposites are summarized in Table 2. All the samples exhibited spherical and polyhedral closed cell structures. Incorporating nanoparticles into the polyurethane matrix effectively reduced the average cell size by promoting nucleation [4,20]. The average cell size of the RPUF nanocomposites decreased below that of neat RPUF as the GNP content increased. This indicates that the GNPs could act as a nucleating agent for bubble formation and growth during foaming. RPUF nanocomposites containing 0.1 wt% (G 0.1) did not show uniform cell morphology. However, a further increase in the GNP loading decreased the cell size, resulting in uniform cell morphology, probably due to stable nucleation of the liquid crystalline GNP. It can be explained that GNP incorporation may provide a physical barrier to cell growth, inhibiting cell coalescence and thus reducing the average cell size of the foams due to the oriented liquid crystalline dispersion.

The closed cell contents of neat RPUF and RPUF nanocomposites with different GNP contents are presented in Figure 8. A higher closed cell content of the RPUF improves its thermal insulation properties by preventing the diffusion of trapped blowing gas from the cells. The closed cell content of the RPUF nanocomposite with 0.1 wt% GNPs was slightly lower than that of the neat RPUF. This was attributed to the presence of partially broken cells. Subsequently, the closed cell contents of the RPUF nanocomposites tended to increase with the GNP content, e.g., up to 93.6% for G 0.4. GNPs with a liquid crystalline orientation during foaming can effectively promote nucleation without cell opening.

Figure 9 presents the thermal conductivity of RPUF nanocomposites with different GNP contents. Typically, the thermal conductivity of polymer nanocomposites containing graphene was expected to increase because of the outstanding thermal conductivity of graphene [3,16,53]. Although graphene has high thermal conductivity, a slight decrease in the thermal conductivity of the RPUF nanocomposites was observed in the present study. The decrease in thermal conductivity of RPUF is one of the most important properties to apply the insulation materials. RPUF nanocomposites reinforced with randomly oriented GNP formed a non-uniform cellular morphology during cell formation, resulting in a slight increase in the thermal conductivity of G 0.1. However, for subsequent GNP concentrations, the thermal conductivity of GNP/RPUF nanocomposites decreased linearly. This was mainly due to the stable nucleation of GNPs with a liquid crystalline orientation, resulting in the formation of smaller cells, as shown in the SEM images. Thus, the decreased cell sizes and increased closed cell contents contributed to the decreased thermal conductivity of the GNP/RPUF nanocomposites. As mentioned above, the thermal conductivity of cellular foam is determined by several parameters, including the gas and matrix conductivity, and thermal radiation. In particular, the reduced average cell size helped to reduce the thermal conductivity by radiation and thus decreased the total thermal conductivity of the resulting nanocomposites [7,9]. Additionally, the uniformly dispersed GNPs in the foam cell walls appeared to act as barriers to radiant heat transfer. This also decreased the radiative conductivity, thus reducing the thermal conductivity of the GNP-containing nanocomposites. Another possible reason for the reduction in thermal conductivity was the reflection or absorption of radiant heat by GNPs with a high surface area [10,54]. In the study on the preparation of extruded polystyrene (PS) with carbon particles, including graphite, activated carbon, and carbon nanofibers, PS/carbon particle foams, especially graphite, showed a significant decrease in thermal conductivity [54]. The lower thermal conductivity of PS/carbon particle foams was due to the strong absorption and reflection of infrared (IR) radiation by carbon particles as well as a reduction in the cell size. Thus, the high absorption and reflection capacity of GNPs with a high surface area also could decrease the thermal conductivity of GNP/RPUF nanocomposites.

Typically, the compressive strength of RPUFs is strongly dependent on their density. To eliminate foam density effects, the compressive strengths of neat RPUF and GNP/RPUF nanocomposites were normalized using Equation (2) [3]:σ_n_ = σ(40/*ρ*)^2^ [{1+√(40/*ρ*_s_)}/{1+√(*ρ*/*ρ*_s_)}]^2^(2)
where σ_n_ represents the normalized compressive strength; σ represents the compressive strength measured in this experiment; *ρ* represents the density of RPUF (kg/m^3^); and *ρ_s_* represents the density of the solid matrix, which was 1200 kg/m^3^. Figure 10 presents the normalized compressive strengths of GNP/RPUF nanocomposites at 10% strain. The enhancement in the compressive strength of RPUF nanocomposites was reported by the incorporation of nano-filler [3,14,15]. All the nanocomposites showed enhanced compressive strength compared with neat RPUF, i.e., 0.241, 0.275, 0.302, 0.299, and 0.291 MPa for control, G 0.1, G 0.2, G 0.3, and G 0.4, respectively. The compressive strengths of the GNP/RPUF nanocomposites were increased with increasing GNP content and leveled off above 0.2 wt% (G 0.2). The enhancement in the compressive strength of the GNP/RPUF nanocomposites was attributed to the reinforcement effects of the nano-fillers. GNPs dispersed in a polymer matrix increase the rigidity of the solid matrix and allows it to withstand more compressive force. Furthermore, uniform dispersion and nucleation of GNPs during foaming contributed to the reduction of the average cell size of the nanocomposites, which also resulted in the enhanced compressive strength of the RPUF nanocomposites.

Figure 11 presents the storage and loss moduli of the RPUF and RPUF nanocomposites as a function of temperature. The storage moduli for all the samples show a plateau region at lower temperatures before decreasing gradually between 120 and 210 °C, corresponding to the transition from glassy to rubbery. Among the samples, G 0.1 showed a lower storage modulus over almost the whole temperature range due to the presence of partially broken cells. No significant improvements were observed in the storage moduli of the GNP/RPUF nanocomposites due to their small GNP contents, with the exception of the nanocomposite containing 0.4 wt% GNPs, which showed an increase in the storage modulus. Notably, the higher storage modulus of G 0.4 over the whole temperature region indicates the enhanced thermal resistance of this GNP/RPUF nanocomposite. In the present study, the glass transition temperatures (T_g_) of neat RPUF and GNP/RPUF nanocomposites were obtained from the maximum points of the loss modulus curves. The glass transition temperature of the GNP/RPUF nanocomposites containing 0.4 wt% of GNPs increased to 148.8 °C compared with 143.7 °C for the neat RPUF, because GNPs with high stiffness might affect the mobility of the polymer chain. The GNP/RPUF nanocomposites containing GNP contents below 0.4 wt% showed no marked enhancement resulting from the GNPs, with all the samples having similar T_g_ values around 141.2 ± 1.5 °C. Incorporating a small GNP content had the possibility of the improvement on thermal insulation without significant deterioration of the mechanical properties of the GNP/RPUF nanocomposites compared with neat RPUF.

## 4. Conclusions

GNP/pMDI dispersions with few defects were obtained by employing a simple and eco-friendly method in the absence of organic solvent under ultrasonication, and GNP/pMDI dispersions with different GNP concentrations were used to prepare RPUF nanocomposites. We focused on the relation between the orientation of GNPs with the layer-by-layer structure of the GNP/pMDI dispersion and the physical properties of the RPUF/GNP nanocomposites. The compressive strength, the closed cell content, and thermal conductivity of the RPUF nanocomposites reinforced with GNPs were closely related to the liquid crystalline behavior of the GNP/pMDI dispersions. Despite the high thermal conductivity of the GNPs, the thermal conductivity of the GNP/RPUF nanocomposites decreased as the GNP concentration increased above the specific concentration of GNPs, showing liquid crystal ordering of the GNPs rheologically. It was found that well-dispersed GNPs in a polymer matrix reduced the radiative thermal conductivity by acting as a barrier to radiant heat transfer, which decreased the thermal conductivity of the GNP/RPUF nanocomposites. Additionally, the GNPs acted as a nucleating agent for bubble formation and growth during foaming, which decreased the average cell size. The decreased cell size and increased closed cell content also lowered the thermal conductivity of the GNP/RPUF nanocomposites. Consequently, the lower thermal conductivity of the GNP/RPUF nanocomposites was closely related to the orientation of the nanofiller in the dispersion as well as the cell size and closed cell content. The strategy to prepare nanocomposites of RPUF investigated in this study can be applied industrially to improve the performance of eco-friendly thermal insulation foams.

## Figures and Tables

**Figure 1 nanomaterials-12-00685-f001:**
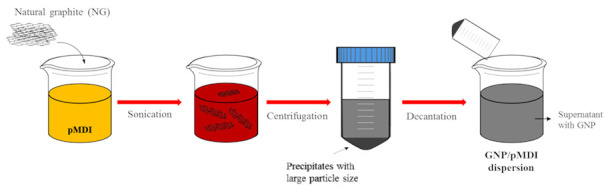
Schematic illustrating the preparation of GNP/pMDI dispersion.

**Figure 2 nanomaterials-12-00685-f002:**
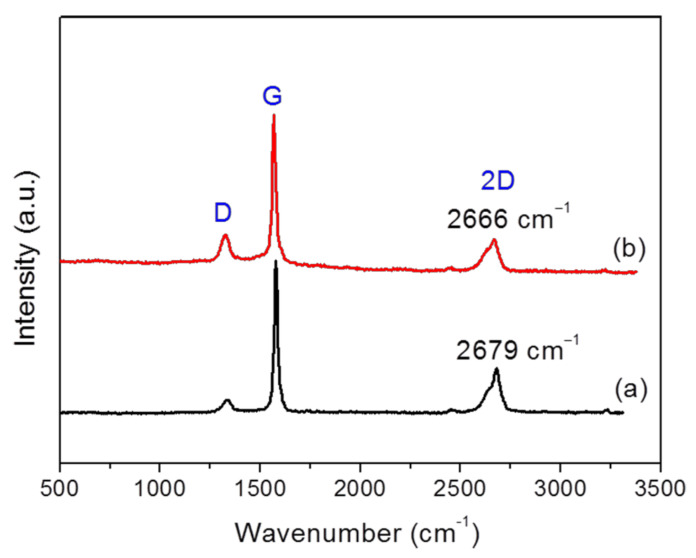
Raman spectra of (**a**) natural graphite (*I_D_*/*I_G_* = 0.1), and (**b**) GNP exfoliated with pMDI (*I_D_*/*I_G_* = 0.2).

**Figure 3 nanomaterials-12-00685-f003:**
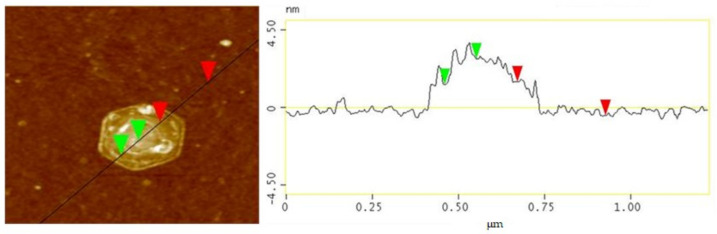
AFM images and line profile of exfoliated GNP with pMDI by ultrasound sonication.

**Figure 4 nanomaterials-12-00685-f004:**
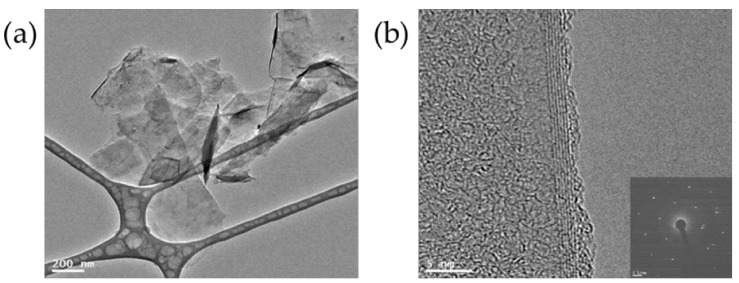
(**a**) TEM image and (**b**) folded edge image of GNP obtained by exfoliation of NG via ultrasound sonication in pMDI (SAED image is given in bottom right). The accelerating voltage to obtain the TEM images was 100 kV.

**Figure 5 nanomaterials-12-00685-f005:**
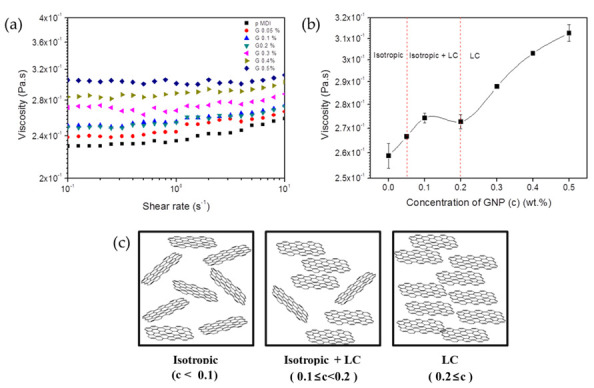
(**a**) Steady shear viscosities of GNP/pMDI master batch with different GNP concentration, (**b**) viscosity versus concentration for GNP/pMDI dispersion at a shear rate of 10/s, and (**c**) schematic illustration on the orientation of GNP in pMDI.

**Figure 6 nanomaterials-12-00685-f006:**
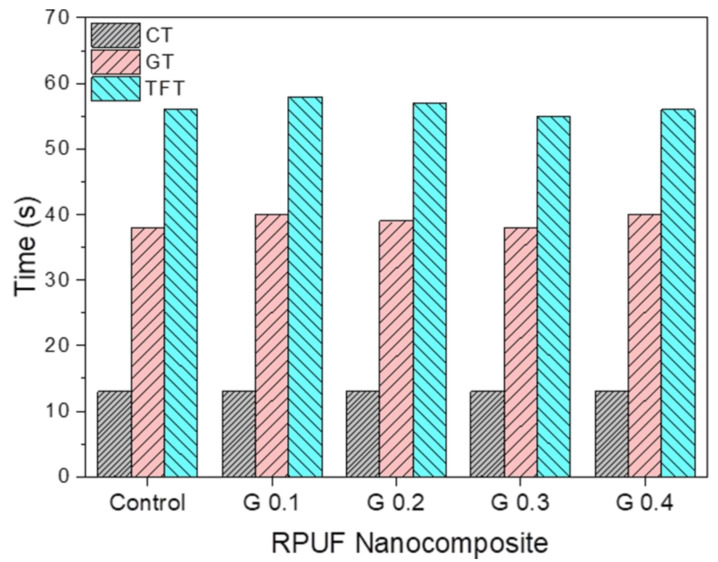
Reaction parameters (cream time, gel time, and tack free time) in foam formulations with different GNP contents.

**Figure 7 nanomaterials-12-00685-f007:**
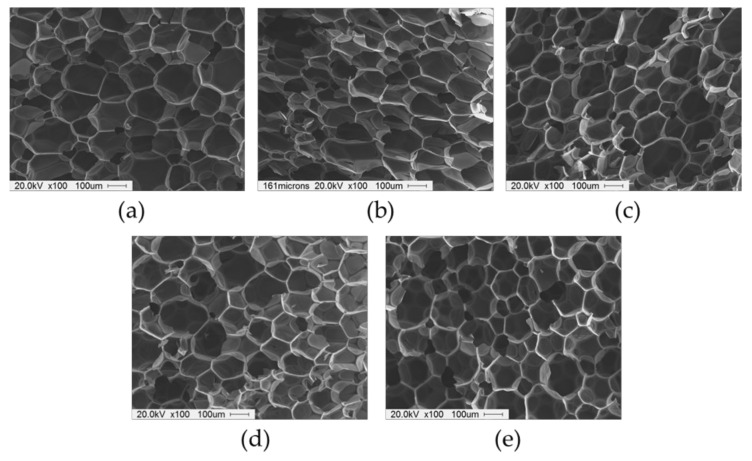
SEM micrographs of RPUF nanocomposites filled with different GNP content: (**a**) control; (**b**) G 0.1; (**c**) G 0.2; (**d**) G 0.3; (**e**) G 0.4.

**Figure 8 nanomaterials-12-00685-f008:**
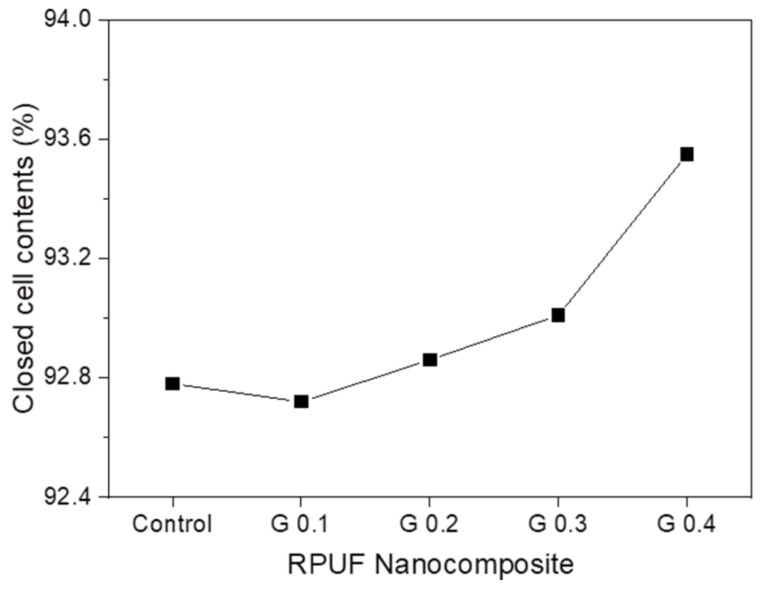
Closed cell content of RPUF nanocomposites with different GNP contents.

**Figure 9 nanomaterials-12-00685-f009:**
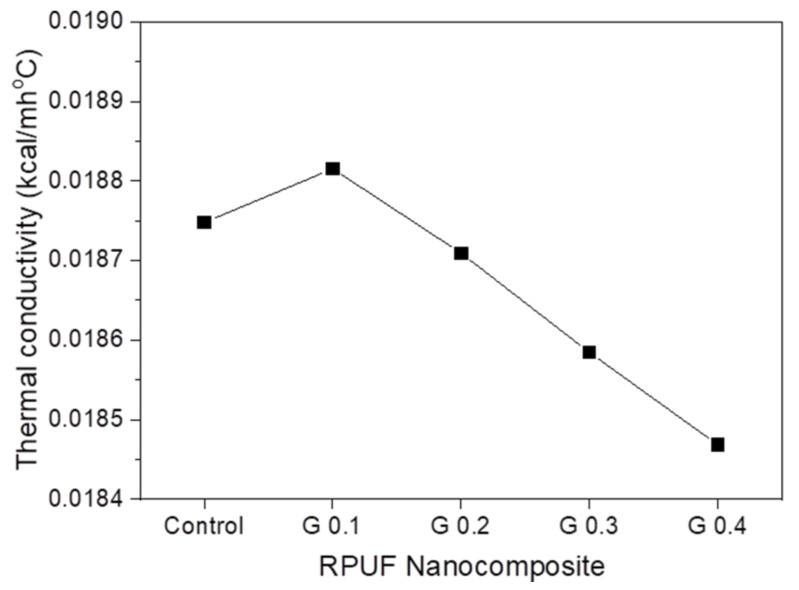
Thermal conductivity of RPUF nanocomposites with different GNP contents.

**Figure 10 nanomaterials-12-00685-f010:**
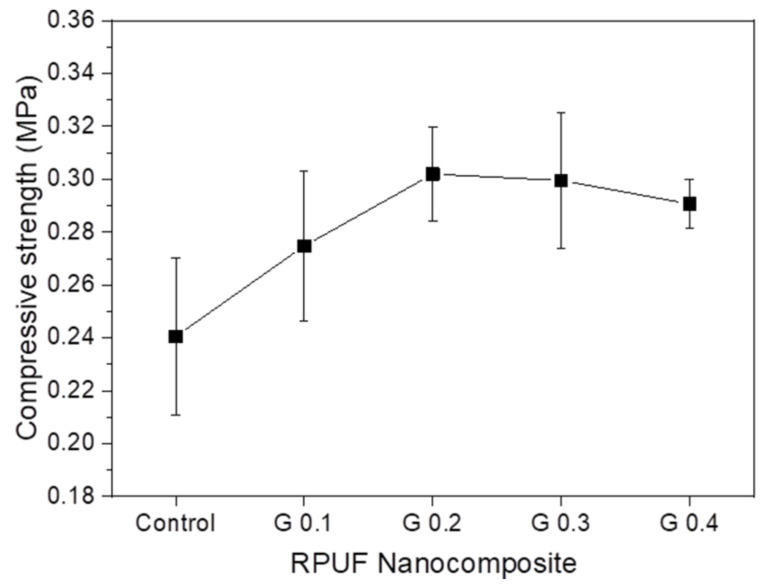
Normalized compressive strength of RPUF nanocomposites with different GNP contents.

**Figure 11 nanomaterials-12-00685-f011:**
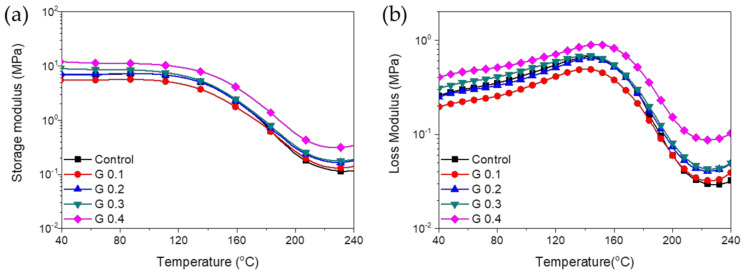
(**a**) Storage modulus and (**b**) loss modulus of RPUF nanocomposites with different GNP content: (□) control; (○) G 0.1; (△) G 0.2 ; (▽) G 0.3 ; (◇) G 0.4.

**Table 1 nanomaterials-12-00685-t001:** Sample codes and formulations of RPUF nanocomposites with different content of GNP.

Sample Code	Feed Composition (g)				
	Control	G 0.1	G 0.2	G 0.3	G 0.4
A component					
Polyol	100.0	100.0	100.0	100.0	100.0
Silicone surfactant	2.0	2.0	2.0	2.0	2.0
PEH	0.3	0.3	0.3	0.3	0.3
DMCHA	1.5	1.5	1.5	1.5	1.5
TCPP	15.0	15.0	15.0	15.0	15.0
Distilled water	0.9	0.9	0.9	0.9	0.9
Physical blowing agent	33.0	33.0	33.0	33.0	33.0
B component					
GNP/pMDI dispersion	105.9	106.0	106.1	106.2	106.3
GNP content in pMDI	0	0.11	0.21	0.32	0.43

**Table 2 nanomaterials-12-00685-t002:** Average cell size, average thickness of cell walls, and number of cells per unit area of FPUFs with different GNP contents.

Sample Code	Control	G 0.1	G 0.2	G 0.3	G 0.4
Average cell size (μm)	204 ± 54	199 ± 69	194 ± 46	180 ± 40	148 ± 43
Number of cells (cm^−2^)	3837	3916	4501	4723	4945

## Data Availability

The data presented in this study are available on request from the corresponding author.

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
