# Peer review of "Nanocomposites of Rigid Polyurethane Foam and Graphene Nanoplates Obtained by Exfoliation of Natural Graphite in Polymeric 4,4′-Diphenylmethane Diisocyanate"

_nanomaterials, 2022, doi:10.3390/nano12040685_

Round 1

Reviewer 1 Report

The paper describes the   effect of graphene nanoparticles on the mechanical and thermal properties of rigid plyurethane foams.  The manuscript is usually clear and well organized and needs minor revision work to be suitable for publication in Nanomaterials. In the following my detailed comments:

  • Page 4, table 1. Units are completely missing. Please add units where necessary.
  • Page 6. Observing Figure 1 it is possible to see a D peak in the starting graphite, which is not usually observed in HOPG graphite. Therefore, the presence of an increased D peak in the exfoliated graphene is not surprising, with a lower quality of the graphene. Some comment will increase the clarity of the report.
  • Figure captions are usually extremely short. A more detailed caption help the reader. In particular in the caption of Figure 4 are not reported at all the measurement parameters such as accelerating voltage and current which is an important parameter in TEM imaging.
  • Page 9, Table 2. There is an inconsistency in the data reported. The number of cell per unit surface of G0.1 is lower than the Control one, with a lower cell dimension. Please check.

After these revisions, I could recommend the publication of this manuscript in Nanomaterials.

Author Response

A PDF file replying to Reviewer 1 was attached.

Reviewer 2 Report

Please find attached my comments.

Author Response

A PDF file replying to reviewer 2 was attached.

Reviewer 3 Report

This manuscript reports the dispersion of graphene in rigid poly(urethane) foams to improve the thermal properties of these materials. The compositions generated are referred to as "nanocomposites" although no good evidence is presented to support this (in fact, modulus is not improved in some cases by the presence of graphene). How is it known that these are not simple filled polymers? This needs to be clarified either by providing some evidence or referring to previous reports which establish the formation of graphene/poly(urethane) composite structures.

The manuscript will require revision for clarity and readability prior to publication. Corrections are penciled-in directly on pages of the manuscript attached. These are illustrative of the kinds of changes needed throughout. In rewriting, careful attention should be paid to the use of articles, tenses and proper sentence structure. Extraneous phrases such as "to the best of our knowledge" should be avoided. A figure cannot "show"; particular properties may be displayed in a figure. Author's names, et. al., and personal pronouns should be omitted.

Author Response

A PDF file replying to reviewer 3 was attached.

Reviewer 4 Report

The manuscript " Nanocomposites of rigid polyurethane foam and graphene nanoplates obtained by exfoliation of natural graphite in polymeric 4,4'-diphenylmethane diisocyanate" fits in  the rigors of your journal and I recommend its publication. I make the following recommendations:

  1. The references must  be updated.
  2. A list with abbreviation would  be very necessary to make the entire text more easy readable.
  3. The practical implications of the research must be emphasized.
  4. The expression "it is postulated" used sometimes in the text is not suitable and the corresponding sentences must be reformulated.

Author Response

A PDF file replying to reviewer 4 was attached.

Round 2

Reviewer 3 Report

This manuscript is much improved. However, the abstract opens with an incomplete sentence. Composite formation has not been demonstrated but has been addressed.

Author Response

Comments are highly appreciated and answers to the comments are attached.
